# Data Pre-Processing and Signal Analysis of Tianwen-1 Rover Penetrating Radar

**Shuning Liu** [1,2]**, Yan Su** [1,2,*]**, Bin Zhou** [3]**, Shun Dai** [1]**, Wei Yan** [1]**, Yuxi Li** [3]**, Zongyu Zhang** [1,2] **, Wei Du** [1,2]
**and Chunlai Li** [1,2]

1 Key Laboratory of Lunar and Deep Space Exploration, National Astronomical Observatories, Chinese Academy of Sciences, Beijing 100101, China
2 School of Astronomy and Space Science, University of Chinese Academy of Sciences, Beijing 100049, China
3 Aerospace Information Research Institute, Chinese Academy of Sciences, Beijing 100094, China
* Correspondence: suyan@nao.cas.cn

**Abstract:** The Rover-mounted Subsurface Penetrating Radar (RoSPR) is one of the scientific payloads onboard China's first independent Mars exploration mission, Tianwen-1. The radar aims to characterize the thickness of the upper Martian soil and investigate the subsurface stratigraphy by collecting and processing the data. This article is mainly divided into two parts, the introduction of data pre-processing and analysis of pre-processed radar signals, aiming at helping scientists make more effective use of radar data. The first part describes the operating principle of the RoSPR and the procedure of radar data pre-processing at all levels. Data pre-processing is mainly designed to transfer the raw data format to a common PDS (Planetary Data System) and eliminate the influence of the instrument. In the signal analysis part, the performances of both self-check signals and echo signals of low- and high-frequency channels are analyzed, which indicate a stable radar system and are useful for background removal. Phase and time calibration is of great importance for improving data quality and making the radar data more accurate. Moreover, further processing is required to obtain clear radar images, such as filtering, background removal and gain setting.

**Keywords:** Tianwen-1; subsurface penetrating radar; Mars; data products; data pre-processing; subsurface penetrating radar





## 1. Introduction

Mars is the fourth planet in the solar system from the inside out. It is a terrestrial planet and has the highest similarity with the Earth in the solar system. With the development of aerospace technology and the diversification of deep space exploration methods, the enthusiasm of various countries for Mars exploration has gradually increased. Some evidence suggests that early Mars was warm, wet and over its history it has experienced significant geological changes caused by surface and subsurface water on Mars [1]. Exploring Mars, searching for life information on Mars and exploring the habitability of Mars have gradually become the mainstream of international deep space exploration. Moreover, rover radar has been successfully used in lunar exploration due to its unique penetrating characteristics [2]; so, radar is also an attractive and powerful technology for Mars exploration.

In 1638, Francisco Fontana succeeded in sketching two general pictures of Mars, which were provided by his telescopic observations [3]. This could have been the first study of Mars in history. Since the 1960s, more than 40 probes have been launched to Mars [4]. In these missions, several of them reached Mars successfully, equipped with scientific instruments, including X-ray spectrometers and cameras. In the past twenty years, Mars Express, MRO (Mars Reconnaissance Orbiter) and other missions of various countries have been making great efforts to search for the existence of groundwater and life on Mars by radar [5]. MARSIS (Mars Advanced Radar for Subsurface and Ionosphere Sounding), an orbiting radar, first spotted the evidence of liquid water trapped below the ice of the

South Polar Layered Deposits. From 2020, ground penetrating radar (GPR) instruments mounted on Mars rovers are increasingly attractive owing to their ability to make in situ measurements on the Martian surface. NASA's Mars 2020 rover carried a surface imaging radar RIMFAX (Radar Imager for Mars' Subsurface Experiment) [6]. It can provide images of subsurface structures and help researchers study the composition of subsurface materials. The RIMFAX is designed to penetrate maximum depths of 10 to 500 m, depending on ground conditions. In addition, the ESA's ExoMars Rover mission, whose objectives are to reveal the geological history of Mars and to search for evidence of life on the planet, will be equipped with a radar named WISDOM (The Water Ice Subsurface Deposit Observation on Mars) [7]. WISDOM is designed to detect the first 3 m of the Martian subsurface with a vertical resolution of a few centimeters [8].

Tianwen-1, China's first independent Mars exploration mission, was launched on 23 July 2020. Moreover, on 15 May 2021, the Zhurong rover successfully made a soft landing on the Utopia Plain. One of the scientific payloads is the Mars Rover Subsurface Penetrating Radar (RoSPR). The main objective of the radar is to characterize the thickness of the upper Martian soil and investigate the depth distribution of the subsurface stratigraphy [4]. Additionally, it is designed to explore the potential buried water ice distribution [9].

As shown in Figure 1, from 25 May 2021 to 15 May 2022, the Zhurong rover moved 1921 m. It walked an average of 17 m per Earth day. In June, July and August 2021, the rover traveled the longest distance, for around 1 km. In contrast, it only moved a total of 186 m in September, October and November 2021. Especially, the rover only operated for two days in October. However, at present, the Mars rover's inspection area has entered winter. Therefore, to cope with the reduction of the solar wing's power generation capacity caused by sand and dust weather, and the extremely low ambient temperature in winter, the Mars rover was set to sleep mode on May 18. It is expected to resume normal work around December this year after the environmental conditions improve.

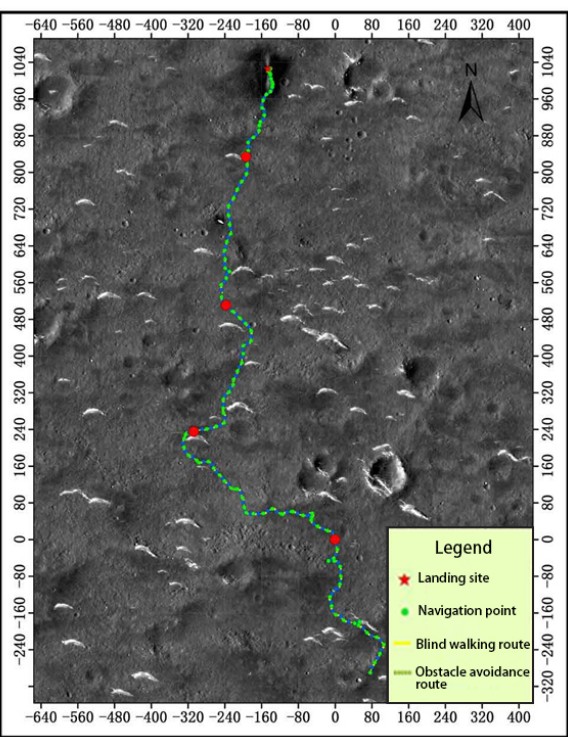

**Figure 1.** Walking track of Mars rover from 25 May 2021 to 15 May 2022. The unit of the coordinate system is meter. The origin is the fifth red star on the walking track from north to south.

## 2. Mars Rover Subsurface Penetrating Radar

Compared with RIMFAX and WISDOM, RoSPR has two operating channels [10]. One works in the frequency range of 15 and 95 MHz, and the other operates from 450 to 2150 MHz. As is displayed in Figure 2, the two low-frequency channel (CH1) antennas are separately mounted on two bottom sides of the top board of the rover, while the two high-frequency channel (CH2) antennas are mounted on the front board of the rover [11]. The design of the CH1 antennas is inherited from the CH1 antennas of the LPR (Lunar Penetrating Radar) onboard the Chang'e-3 Mission [12]. In Figure 2c, the CH1 transmitting and receiving antennas are spaced about 1034 mm apart. The low-frequency channel uses linear frequency modulation (LFM) signals and has a pair of monopole antennas whose length is 1350 mm and diameter is 12 mm. It has only one polarization and can probe the Mars ground down to 100 m depth with a vertical resolution of a few meters. The high-frequency channel uses frequency-modulated interrupted continuous wave (FMICW) signals and contains two sets of orthogonally polarized antennas based on Vivaldi design—one for transmitting and one for receiving [13]. The spacing between CH2 antennas is 420 mm in Figure 2c. Moreover, the transmitting antenna and the receiving antenna are installed on the front side panel of the rover with a center interval of 500 mm. HH (Horizontal transmit, Horizontal receive), HV (Horizontal transmit, Vertical receive), VH (Vertical transmit, Horizontal receive), and VV (Vertical transmit, Vertical receive) full-polarization detection can be realized through the configuration of H-polarized and V-polarized transceiver antennas.

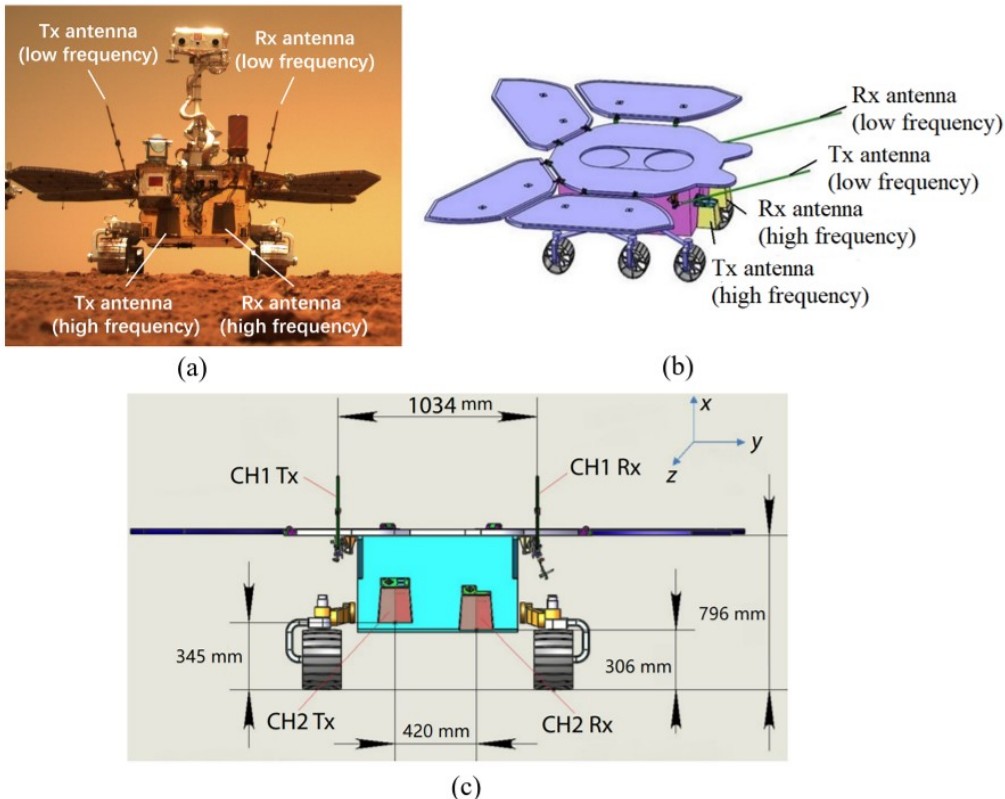

**Figure 2.** Images of Mars Rover. (**a**) Zhurong selfie, taken by the deployable Tianwen-1 Remote Camera. (**b**) Schematic diagram of Zhurong rover and RoSPR antennas on it. (**c**) Detailed positions of the RoSPR antennas on the rover.

Table 1 presents the main characteristics of the RoSPR. The low-frequency channel, which has only one polarization mode, is designed to penetrate depths of 10 to 100 m with a resolution of a few meters. In contrast, the high-frequency channel, which has four

different polarization modes, can penetrate a maximum depth of 3 to 10 m with a better resolution of a few centimeters.

**Table 1.** The main characteristics of the RoSPR [4].

| Parameter | Value | |
| --- | --- | --- |
| | CH1 | CH2 |
| Frequency range | 15~95 MHz | 0.45~2.15 GHz |
| Center frequency | 55 MHz | 1.45 GHz |
| Channel polarization | HH | HH, HV, VH, VV |
| Depth resolution | A few meters | A few centimeters |
| Penetrating Depth | 100 m (ice) 10 m (soil) | 10 m (ice) 3 m (soil) |

As the rover walks, the radar transmits electromagnetic signals to the Martian underground, propagating through the Martian subsurface medium. When encountering some special targets such as soil layers, ice layers and rocks, electromagnetic waves will be reflected and scattered to form radar echoes. By receiving and processing radar echoes, target information with discontinuous electromagnetic parameters can be acquired, and cross-sectional images of geological layers, rocks and possible ice sheets underlying the rover's path can be formed.

The RoSPR has three working modes, including standby mode, detection mode and power-on self-check mode, as shown in Table 2.

**Table 2.** Three working modes of the Mars Rover Subsurface Penetrating Radar

| Working Mode | Definition | Working Condition |
| --- | --- | --- |
| Standby mode | The radar is powered up, receiving parameters and command injection, and sending engineering and telemetry parameters. There is no radar signal transmission and scientific data generation in this mode. | The default mode after the power is on, and the system works in the minimum power consumption state. |
| Detection mode | The radar transmits radar signals for detection and receives echo signals. | Carry out continuous detection when the Zhurong rover moves. |
| Self-check mode | No radar signal is sent, and only signal reception is performed. | Before the detection mode. |

It is worth noting that there are two working ways in the detection mode, including the fixed distance trigger working mode and the timed trigger working mode. For the fixed distance trigger working mode, the detection is carried out according to the fixed distance spacing. The trace spacing of the low-frequency channel can be adjusted in multiple steps such as 50 cm and 25 cm, and the trace spacing of the high-frequency channel can be adjusted for the steps such as 20 cm and 10 cm. In practice, the CH1 trace spacing was adjusted to 50 cm before 17 August 2021; then, from that day until now, it has been set to 25 cm. The trace spacing of the high-frequency channel is 5 cm. For the timing trigger working mode, the detection is performed at fixed time intervals determined by data injection.

### 3. Data Pre-Processing

The scientific raw data are achieved by the ground stations of the National Astronomical Observatories, Chinese Academy of Sciences. One of the objectives of the Ground Research and Application System (GRAS) is data acquisition and pre-processing. In order

to obtain high-quality radar data, data pre-processing is necessary to eliminate the influence of the instrument.

The Data Pre-processing Subsystem (DPS) is a segment of GRAS. Similar to the Chang'e-3 mission, the task of DPS is pre-processing of raw data collected by RoSPR, including package sorting, identification of geographical location, calculation of probe azimuth angle, probe zenith angle, solar azimuth angle and so on [14]. Therefore, these processes produce three levels of data products, including Level 0, Level 1 and Level 2 data. Additionally, the data levels of data products are defined according to NASA and CODMAC processing levels for PDS (Planetary Data System) datasets. The data are usually categorized into three levels, Level 0, Level 1 and Level 2. Level 0 is further divided into Level 0A and Level 0B. Level 0 and Level 1 processing are applied to all instruments onboard, but Level 2 processing varies among different instruments. Figure 3 shows a flow chart of pre-processing for different data levels of RoSPR.

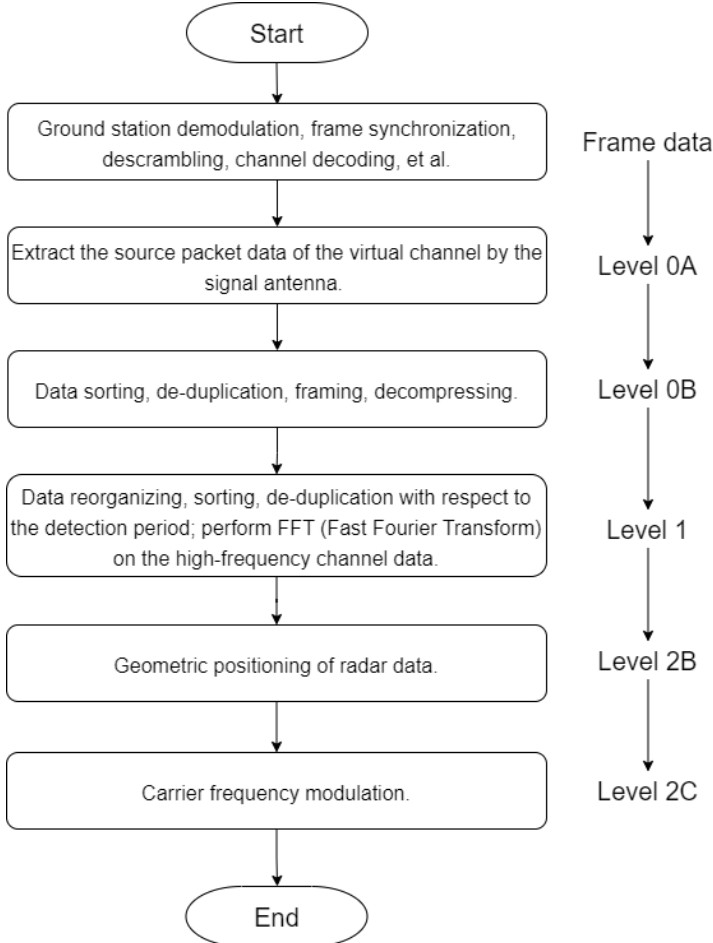

**Figure 3.** Flow chart of scientific data pre-processing for RoSPR.

### 3.1. Frame Data

When the ground station receives the detection data of scientific payloads, it needs to demodulate, synchronize and decode the signal, and convert the raw data into frame data. Therefore, the frame data can be treated as raw data, and the following steps are based on processing the frame data.

### 3.2. Level 0 Data

Level 0 data processing is divided into 0A data and 0B data processing.

### 3.2.1. Level 0A Data

Level 0A data processing consists of two steps. The first step is to read frame data through frame synchronization. According to the frame data format, extract the frame data whose spacecraft sign is $0 \times 52$ and read the 896-byte length VCDU (Virtual Channel Data Unit) data frame of the rover. Then, according to the first 64 bytes of the frame header information, determine the quality of the frame data and write the corresponding quality information bit. The second step is to perform VCDU data frame synchronization. According to the VCDU data frame format and the virtual channel identification, the channel data of the rover scientific load are extracted and formed. At the same time, the statistical information of the data virtual channel is recorded and written into the channel processing report. Finally, add the quality information to each data frame to form the level 0A data.

### 3.2.2. Level 0B Data

The primary purpose of the level 0B data processing is to extract and generate the source package data of RoSPR from the level 0A data, which is an application-oriented protocol standard data unit, including the packaging protocol data unit and the bitstream protocol data unit.

- Sorting and deduplication of data received by multiple antennas.

According to the time code and the virtual channel frame count, sorting and deduplicating of the received level 0A data were performed. Moreover, the duplicated data with the best quality are retained based on the principle of quality priority (the smaller the value, the better the quality). It should be noted that in the case of the same quality, the data collected by the receiving antenna with better performance are retained.

- Extracting the source package of the payload.

First, the source package is synchronized according to the synchronization code (E225H). Then, the CH1 and CH2 scientific data source packages of the RoSPR are extracted, respectively, based on the application process identifier. Moreover, the level 0A data quality is written into the level 0B quality information report. Additionally, according to the time code and the package sequence count, the CH1 and CH2 scientific data source packages mentioned above are sorted and deduplicated, respectively.

- Scientific data framing.

First of all, according to the scientific data frame format of the RoSPR, the data frame is extracted. Moreover, a CRC (Cyclic Redundancy Check) check is carried out for each scientific data frame, and the corresponding result is written into the quality bit of level 0B data. Then, the low-frequency channel and high-frequency data frames are extracted and checked as to whether their data are compressed. If compressed, the scientific data should be decompressed. After decompression, the channel data of each point are stored by 20 bit and their content before decompression is replaced. Meanwhile, to form the level 0B data format, the values of the length of data frames and scientific data are recalculated according to the decompression results. Finally, the scientific data frames of the low-frequency and high-frequency channels are sorted and deduplicated, respectively. Therefore, the final data products include 0B-level scientific data of low-frequency and high-frequency channels. It should be noted that since the scientific data contain the real part and the imaginary part, when transferring data, for the low-frequency channel, the real part is transferred first; however, for the high-frequency channel, the two parts are transferred at the same time.

### 3.3. Level 1 Data

Level 1 data processing is based on level 0B data. The purpose of the processing is to obtain low-frequency and high-frequency data of RoSPR in a detection period and to merge, sort and deduplicate the data. The low-frequency and high-frequency data are processed individually. For low-frequency data, they are stored in two data products. One is the real part data only and the other one is the complex data including the real part

data and imaginary part data. For high-frequency data, since they consist of four different polarization modes, they are stored in four data products.

- High-frequency channel data processing.

For the single polarization raw data of the high-frequency channel, the channel data of different polarization modes are separated. Then, a 4096-point FFT is performed on the scientific data in each trace, and the real and imaginary parts of 2048 points are extracted for storage, where each datum occupies 4 bytes and the length of each scientific data trace is 16,384 bytes.

Similarly, for the unipolar real and imaginary parts of the pulse compression data of the high-frequency channel, the channel data of different polarization modes are separated. When storing the data of each trace, each datum is converted from a 20-bit signed integer to a float-type number and is stored in 4 bytes. Moreover, the length of each effective scientific data trace is 4096 bytes.

- Low-frequency channel data processing.

For the low-frequency data, which only contain the real part, the scientific data of each trace after the detection cycle division are output directly, where each datum is converted from a 20-bit signed integer to a float-type number. It is stored as 4 bytes, and the length of each data trace is 6000 bytes. For the trace data with sampling time windows of 15 us, 10 us, 5 us and 2.5 us, the effective points of each trace are 1500, 1000, 500 and 250, respectively, and the lengths of the data are 6000, 4000, 2000 and 1000 bytes. If the length of the effective scientific data is less than 6000, zeros will be supplemented.

For the low-frequency data, which contains both real and imaginary parts of data, both the output and storage processing are the same. However, since the data have two parts, the length of each data trace is doubled, which is 12,000 bytes. Similarly, with the same sampling time window, the effective points of each trace and the corresponding data length are also doubled.

*3.4. Level 2 Data*

This step has been divided into two parts, including 2B data and 2C data processing. It is worth noting that according to the definition of the scientific data product level of the ground application system, the level 2A data are the product obtained after the calibration results, which are used for correction processing. Since RoSPR does not involve this part of data processing, the level 2A data are ignored in this part.

3.4.1. Level 2B Data

The level 2B scientific data are mainly achieved by adding the information of geometric position based on the level 1 data. By calculating and recording the position and attitude of the rover in the central coordinate system of the Mars surface station at the time corresponding to the detection data, the position and attitude of the rover in the central coordinate system of the lander are transformed. Thus, the geometric position of each detection data will be carried out on the level 2B data.

The input data of this step are the level 1 data and the output data are the level 2B data, corresponding to the PDS data label.

- Geometric positioning of the Mars rover in the lander station center coordinate system.

In this step, three coordinate transformation matrices are constructed, including the matrix from the Mars rover coordinate system to the Mars surface working coordinate system, the Mars surface working coordinate system to the Mars surface station center coordinate system and the Mars surface station center coordinate system to the landing site station center coordinate system. Therefore, the transformation result of the parameters in the landing point station center coordinate system is used as the geometric positioning result.

- Other geometric information processing.

This step is to process the geometric information related to the detection data such as the incident angle of the sun and the azimuth angle of the sun.

- The 2B level data and the corresponding PDS label generating.

According to the PDS format, the relevant geometric information of the rover in the station center coordinate system of the landing site, such as the position, attitude and sun altitude angle, are organized. Then, the corresponding level 2B scientific data products and their corresponding PDS labels are generated and output.

### 3.4.2. Level 2C Data

The level 2C data are achieved by conducting carrier modulation on the level 2B data. By carrier modulation of the low-frequency and high-frequency channel data, respectively, the data frequencies −40 MHz~40 MHz and 0 MHz~1700 MHz of the low-frequency and high-frequency channels of the RoSPR are restored to the operating frequencies of 15~95 MHz and 450 MHz~2150 MHz to realize the original frequency restoration.

- Carrier modulation for low-frequency channel data.

For the low-frequency channel data, a 4 times up-sampling is first performed on the level 2B data. Then, a 55 MHz carrier processing is performed to move the data frequency from −40 MHz~40 MHz to 15~95 MHz. It is worth noting that the up-sampling method is sinc interpolation, and points per record is up-sampled from 1500 to 6000. The corresponding formula is as follows.

$$LF\_Data\_2C\_result = LF\_Data\_2C\_result \times e^{j2\pi \cdot 55 \cdot 10^6}, \tag{1}$$

In this formula, *LF_Data_2C_result* is the level 2C product data, *LF_Data_2B_result* is the level 2C product data, j is the complex number identifier and t is the low-frequency time axis. Carrier modulation for the average low-frequency channel signal is shown in Figure 4a. It is apparent that it carries the frequency range from −40 MHz~40 MHz to 15~95 MHz.

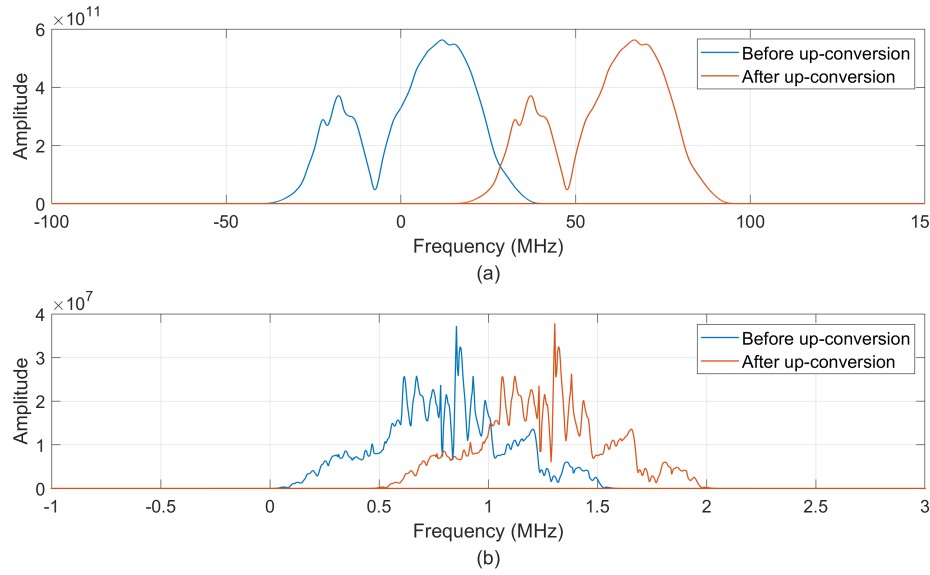

**Figure 4.** Carrier modulation. (**a**) Up-conversion for an average low-frequency channel data. (**b**) Up-conversion for an average low-frequency channel data.

- Carrier modulation for high-frequency channel data.

For the high-frequency channel data, 450 MHz carrier processing is directly performed to move the data frequency from 0 MHz~1700 MHz to 450 MHz~2150 MHz. The corresponding formula is as follows.

$$HF\_Data\_2C\_result = HF\_Data\_2C\_result \times e^{j2\pi \cdot 450 \cdot 10^6}, \tag{2}$$

Similarly, *HF_Data_2C_result* is the level 2C product data, *HF_Data_2B_result* is the level 2C product data, j is the complex number identifier and t is the low-frequency time axis after pulse pressure processing. Carrier modulation for a single high-frequency channel signal is shown in Figure 4b. It is apparent that it carries the frequency range from 0 MHz~1700 MHz to 450 MHz~2150 MHz.

## 4. Final Published Data

The RePoR data are published by the National Astronomical Observatory of Chinese Academy of Sciences and Ground Application System [15].

The final released scientific data are 2C-level scientific data, including the real and imaginary parts of the pulse compression data. The 2C level subsurface penetrating radar data are generated by carrier modulation of low-frequency and high-frequency detection data based on 2B level data. The data adopt the PDS4 (Planetary Data System) format, and each datum contains two paired files: the data file and the corresponding label file. Moreover, the complex data of CH1 and CH2 have different transmitting ways. For the low-frequency channel data, the transmission sequence of each frame is that the real part is transmitted before the imaginary part, which means that they are transmitted in series. For the high-frequency channel data, the real part and the imaginary part are transmitted in parallel.

Table 3 indicates the list of published RoSPR products.

**Table 3.** List of published RoSPR level 2C products.

| Data Product File Name | Storage Format | File Description |
|---|---|---|
| HX1-Ro_GRAS_***_SCI_N_ yyyymmddhh- miss_YYYYMMDDHHMISS_ ob_ver.2C | Binary | Table_Binary |
| HX1-Ro_GRAS_***_SCI_N_ yyyymmddhh- miss_YYYYMMDDHHMISS_ ob_ver.2CL | XML | Data label |
| *** represents RoSPR-LF-R, RoSPR-LF-RI, RoSPR-HF-HH, RoSPR-HF-HV, RoSPR-HF-VH and RoSPR-HF-VV | | |

In Table 3, RoSPR-LF-R, RoSPR-LF-RI, RoSPR-HF-HH, RoSPR-HF-HV, RoSPR-HF-VH and RoSPR-HF-VV represent low-frequency channel real data, low-frequency channel real and imaginary data, high-frequency channel HH polarization data, high-frequency channel HV polarization data, high-frequency channel VH polarization data and high-frequency channel VV polarization data.

Table 4 displays the data format of the published data of each trace for both CH1 and CH2. It lists the parameters in the published data sequentially and indicates their byte order at the same time. It is quite convenient to read the published data file by either Python or MATLAB according to this table.

**Table 4.** The definition of the published radar data, including CH1 and CH2.

| Item | Parameter Name | Bytes | Byte Order |
|---|---|---|---|
| 1 | Channel | 4 | 1∼4 |
| 2 | Time | 6 | 5∼10 |
| 3 | Frame count | 1 | 11 |
| 4 | Data node | 1 | 12 |
| 5 | Sampling window (CH1) Work mode (CH2) | 1 | 13 |
| 6 | Work mode (CH1) Trigger mode (CH2) | 1 | 14 |
| 7 | Trigger mode (CH1) Antenna polarization (CH2) | 1 | 15 |
| 8 | Detection interval (CH1) Power-on self-test switch (CH2) | 1 | 16 |
| 9 | Effective data length | 2 | 17∼18 |
| 10 | Data | 12,000 (CH1) 16,348 (CH2) | 19∼12,018 (CH1) 19∼16,402 (CH2) |
| 11 | Velocity | 6 | 12,019∼12,024 (CH1) 16,403∼16,408 (CH2) |
| 12 | Position X | 2 | 12,025∼12,026 (CH1) 16,409∼16,410 (CH2) |
| 13 | Position Y | 2 | 12,027∼12,028 (CH1) 16,411∼16,412 (CH2) |
| 14 | Position Z | 2 | 12,029∼12,030 (CH1) 16,413∼16,414 (CH2) |
| 15 | Pitch | 2 | 12,031∼12,032 (CH1) 16,415∼16,416 (CH2) |
| 16 | Yaw | 2 | 12,033∼12,034 (CH1) 16,417∼16,418 (CH2) |
| 17 | Roll | 2 | 12,035∼12,036 (CH1) 16,419∼16,420 (CH2) |
| 18 | Rover location X | 4 | 12,037∼12,040 (CH1) 16,421∼16,424 (CH2) |
| 19 | Rover location Y | 4 | 12,041∼12,044 (CH1) 16,425∼16,428 (CH2) |
| 20 | Rover location Z | 4 | 12,045∼12,048 (CH1) 16,429∼16,432 (CH2) |
| 21 | Rover attitude roll | 4 | 12,049∼12,052 (CH1) 16,433∼16,436 (CH2) |
| 22 | Rover attitude pitch | 4 | 12,053∼12,056 (CH1) 16,437∼16,440 (CH2) |
| 23 | Rover attitude yaw | 4 | 12,057∼12,060 (CH1) 16,441∼16,444 (CH2) |
| 24 | Solar incidence angle | 1 | 12,061∼12,064 (CH1) 16,445∼16,448 (CH2) |
| 25 | Solar azimuth angle | 1 | 12,065∼12,068 (CH1) 16,449∼16,452 (CH2) |
| 26 | Quality information | 1 | 12,069 (CH1) 16,453 (CH2) |

## 5. Analysis of Self-Check Signals

Each time before the radar carries out continuous detection, it requires experiencing the self-check mode, where self-check signals will be received. The received data indicate that a self-check is conducted every 3 m that the rover moves on average. Although both the low-frequency and high-frequency channel data consist of self-check signals, they were generated in different ways. For the CH1 channel, both the transmitter and receiver are in operation, and the self-check signal is a loopback signal. This means that the stability of CH1 channel self-check signals may indicate the stability of the radar system to some extent.

Figure 5 displays the normalized energy of CH1 self-check signals from 25 May 2021 to 29 August 2021, a total of 1 km that the rover walked. Although there are several small vibrations, the general trend of this line is straight, which means that the energy of each CH1 self-check signal is similar. Additionally, to make it clearer, the coefficient of variance (CV) of each self-check signal energy, which is the ratio of the standard deviation to the mean, is calculated. The CV measures the extent of variability concerning the mean of the dataset [16]. The higher the CV value, the lower the dispersion of the dataset. The CV of CH1 self-check signal energy is 0.0267. This low value indicates that the dispersion of energy is low, which means that the CH1 self-check signal is relatively stable; further, this shows that the stability of the radar system of CH1 is good. For the CH2 channel, no radar signal is sent and only the receiver works; so, the self-check can represent the noise and interference of the environment and the equipment. Normally, background removal is performed by subtracting the means of all the track signals. Therefore, in this condition, CH2 channel self-check signals can be combined with normal methods of background removal in the further data processing.

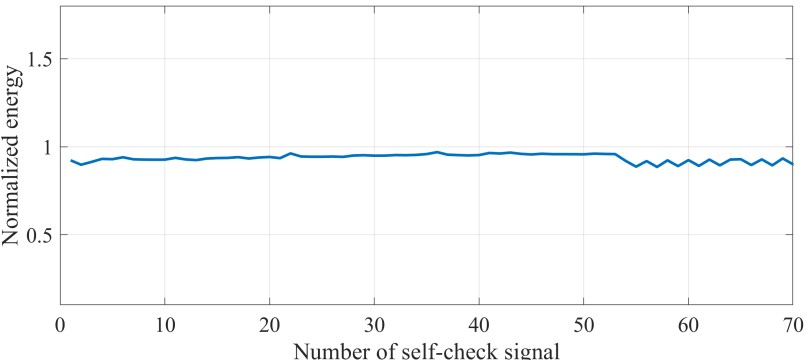

**Figure 5.** Normalized energy of CH1 self-check signals (from 25 May 2021 to 29 August 2021).

Figures 6 and 7 display self-check signal samples of CH1 and CH2, respectively. They both show that the self-check signals of the two channels have great similarities, and they are smaller than normal signals in amplitude. Due to the operating frequency difference, the peak amplitude of CH1 self-check signals occurs at around 190 ns while the peak amplitude of CH2 self-check signals occurs at around 25 ns. This also indicates that the penetrating depth of the low-frequency channel is much deeper than that of the high-frequency channel. Table 5 displays some parameters of CH1 and CH2 self-check signals.

Table 5 shows that the average energy of CH1 self-check signals is much bigger than that of CH2 self-check signals. However, both of their energy is lower than that of their corresponding normal signals. According to the normalized standard deviation, their relatively small value indicates that the dispersion of self-check signal energy is not large, which means that self-check signals of either CH1 or CH2 are similar in energy. Alternatively, the correlation coefficient is close to 1. This also indicates that the self-check signals of the two channels are similar.

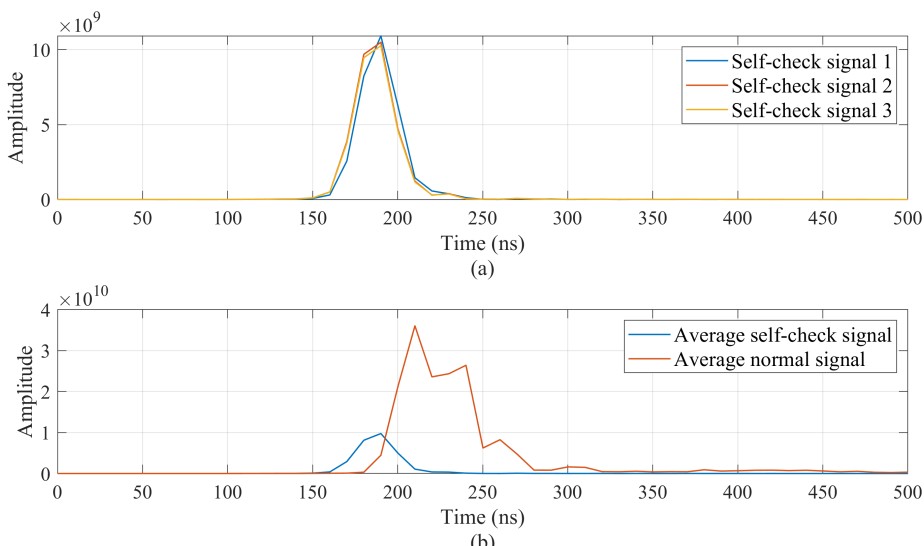

**Figure 6.** (**a**) Three CH1 self-check signal samples. (**b**) CH1 average self-check signal vs. average normal signal.

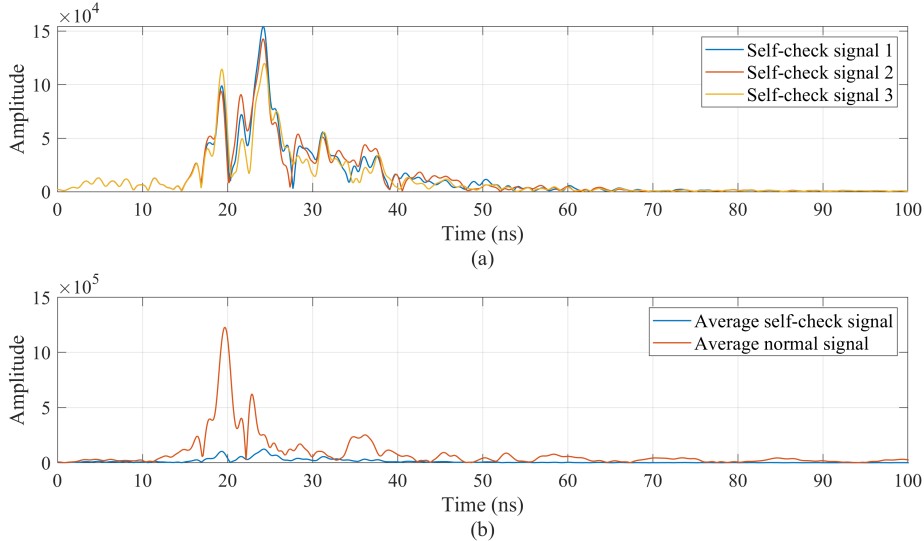

**Figure 7.** (**a**) Three CH2 self-check signal samples. (**b**) CH2 average self-check signal vs. average normal signal.

**Table 5.** Parameters of CH1 and CH2 self-check signals.

|  | Average Energy | Average Energy of Normal Signals | Normalized Energy Standard Deviation | Correlation Coefficient |
| --- | --- | --- | --- | --- |
| **CH1** | $2.38 \times 10^{20}$ | $4.82 \times 10^{21}$ | 0.025 | 0.82 |
| **CH2** | $1.78 \times 10^{11}$ | $3.72 \times 10^{13}$ | 0.136 | 0.83 |

## 6. Calibration of Low-Frequency Channel Data

Although the final published CH1 channel data have been pre-processed, further normal processing such as background removal and filtering cannot be applied to them directly, since there are two types of signals in the low-frequency channel data, as displayed in Figure 8. It is apparent that there are yellow and blue straight lines at around 250 ns displayed in Figure 8e. When the low-frequency channel echo acquisition circuit is powered

on each time, there is a random deviation of a sampling time interval (2.5 ns, corresponding to 400 MHz sampling rate) in the data sequence, so there is a 2.5 ns time deviation and the corresponding 49.5° phase difference in CH1 data, which contributes to two different shapes of waveforms in both the real part and imaginary part of CH1 data. Therefore, time and phase correction are required for further processing of the low-frequency channel data.

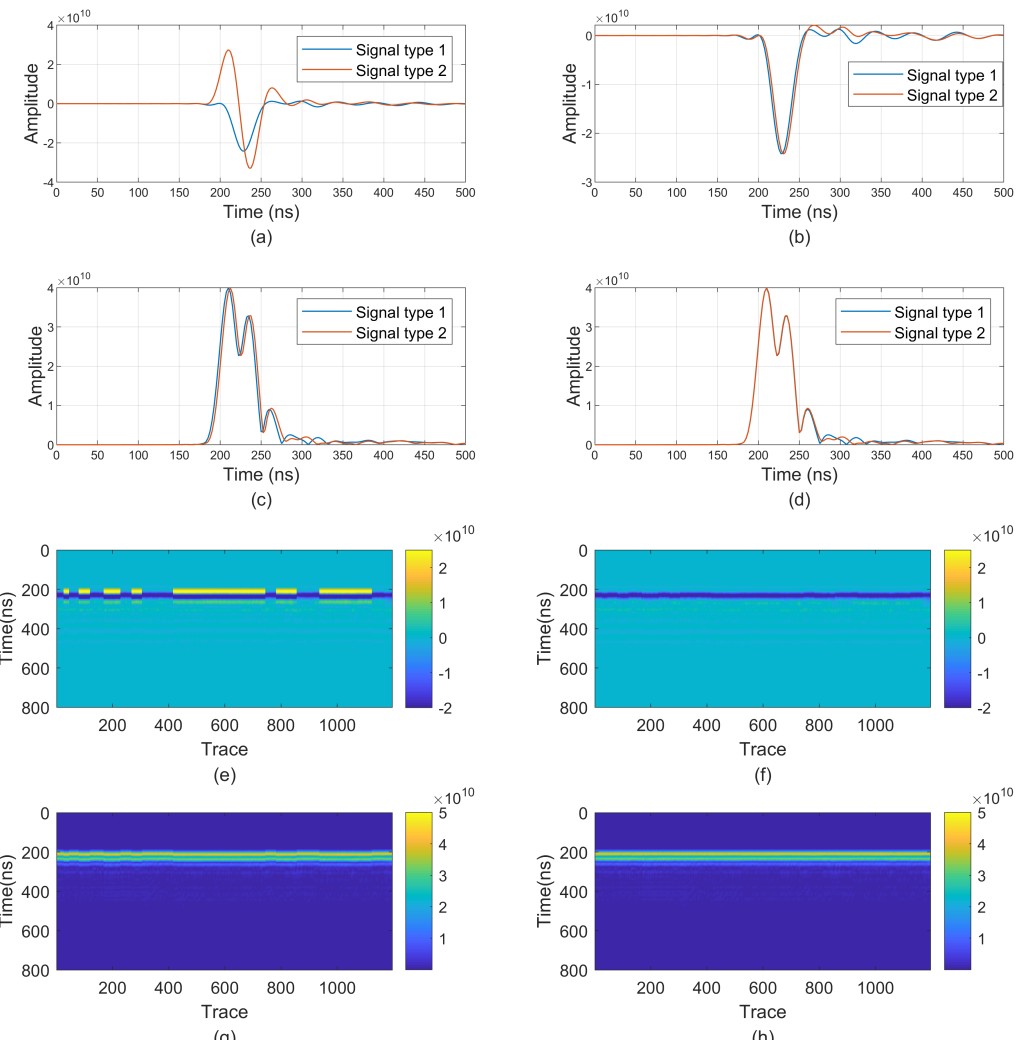

**Figure 8.** A-scans and B-scans of low-frequency channel data. (**a**) A-scan of CH1 real part of data before phase correction; (**b**) A-scan of CH1 real part of data after phase correction; (**c**) A-scan of CH1 absolute value of data before time correction; (**d**) A-scan of CH1 absolute value of data after time correction; (**e**) B-scan of CH1 real part of data; (**f**) B-scan of CH1 real part of data after phase correction; (**g**) B-scan of CH1 absolute value of data; (**h**) B-scan of CH1 absolute value of data after time correction.

For phase correction, signals that have a 49.5° phase difference from normal signals should be calibrated. It should be mentioned that signals displayed in the real or imaginary part of the radar data individually will make it clearer to see the difference between the two types of signals since any changes in either the real part or the imaginary part of the data will make a phase change of this signal. In Figure 8a, the type 1 signal is the normal signal and the type 2 signal is the one that has a phase difference from the signal type 1. They have different shapes in the real part. However, as shown in Figure 8b, after phase correction, the two types of signals tend to be similar. Same as B-scans in Figure 8e,f, after phase correction, two types of signals are presented in dark blue lines instead of yellow and blue lines.

Similarly, in the time domain, a 2.5 ns time difference should be calibrated. Figure 8c displays A-scans of two types of signals in the time domain. It is obvious that signal type 2 requires moving forward 2.5 ns along the time axis to keep consistent with signal type 1. After time correction, their A-scans are consistent in the time domain. Moreover, both the A-scans and B-scans displayed in Figure 8c,d,g,h are presented in absolute values since they can ignore the phase difference and show the time difference more clearly. After time correction, the line with several small vibrations becomes a straight line. Therefore, both time and phase calibration make the two types of CH1 signals the same type of signal.

## 7. Conclusions

Data pre-processing is helpful for researchers to conduct further data analysis, processing and imaging. It transfers the format of the raw data to a more tractable format. However, based on the level 2C scientific radar data, further processing needs to be performed to achieve clear radar images, such as self-check data removal, energy normalization, background removal, etc. Since both the Mars Rover Penetrating Radar and the Lunar Penetrating Radar are Ground Penetrating Radar, they have similar steps in data processing, such as self-check data removal and background removal [2]. However, the processing steps introduced above are not enough. To further improve the resolution and the signal-to-noise ratio of the radar data, more methods such as filtering and gain resetting are necessary [17]. A bandpass filter should be applied to the radar data to eliminate the noise outside the operating frequency bands as much as possible; thus, the signal-to-noise ratio will be improved. In addition, gain resetting aims to amplify a weak radar signal to improve the resolution of the radar image. Here, an SEC gain may be a suitable choice. Moreover, the parameter settings of either bandpass filter or gain are not determined, and they need to be tested several times to achieve a better quality of data.

Since the RoSPR has low- and high-frequency channels with full polarization detection, data collected by these channels after processing play a significant role in detecting water–ice on Mars, which is one of the critical objectives of Mars missions all over the world, including the Tianwen-1 mission.

To sum up, data pre-processing is a significant step to process the raw radar data of good quality for further analysis. It is mainly designed to transfer the format of the raw data to a more common one and eliminate the influence of the instrument. Through the analysis of self-check signals, more information about the radar system could be obtained and it also provides a creative idea for background removal. Additionally, phase and time correction for CH1 data are of great importance, since the calibration is beneficial for further processing, making the radar data more accurate. Based on the pre-processing, further processing needs to be carried out to obtain clear radar images, such as self-check data removal, energy normalization and background removal. Moreover, filtering and gain resetting are also significant for improving data quality, and further research needs to be performed to achieve suitable parameters for filtering and gain resetting.

**Author Contributions:** Conceptualization, S.L., Y.S., B.Z., Y.L. and W.Y.; methodology, S.L., Y.S., B.Z., Y.L. and W.Y.; software, S.L.; resources, B.Z. and Y.L.; data curation, S.L. and Y.S.; writing—original draft preparation, S.L., Z.Z. and W.D.; writing—review and editing, S.L., Y.S., S.D., Z.Z., W.D. and C.L. All authors have read and agreed to the published version of the manuscript.

**Funding:** This research was funded by the National Natural Science Foundation of China (grant number 12073048).

**Acknowledgments:** This work has been supported by the team "Searching for Subglacial Water on Mars with Orbiting Ground Penetrating Radars" of the International Space Science Institute (ISSI).

**Conflicts of Interest:** The authors declare no conflict of interest.

**Abbreviations**

The following abbreviations are used in this manuscript:

| | |
|---|---|
| RoSPR | Mars Rover Subsurface Penetrating Radar |
| GPR | Ground Penetrating Radar |
| MRO | Mars Reconnaissance Orbiter |
| MARSIS | Mars Advanced Radar For subsurface and Ionosphere Sounding |
| RIMFAX | Radar Imager for Mars' Subsurface Experiment |
| WISDOM | The Water Ice Subsurface Deposit Observation on Mars |
| LFM | Linear Frequency Modulation |
| FMICW | Frequency-modulated Interrupted Continuous Wave |
| PDS | Planetary Data System |

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
