# Peer review of "Data Pre-Processing and Signal Analysis of Tianwen-1 Rover Penetrating Radar"

_remotesensing, doi:10.3390/rs15040966_

Round 1

Reviewer 1 Report

Dear authors,

Here you can find the review of the manuscript remotesensing-2086905. The manuscript is conforming to the journal requirements and formats, and a suitable quality data have been included. The major recommendation the authors must be taken into account is related to include a comparison with previously reported works about GPR data pre-processing. Benefits and limitations of the presented data pre-processing are not included. Furthermore, they must be compared to others Penetrating Radar works on the moon or on Mars (NASA missions). Authors are also encouraged to take into account some comments and suggestions.

I hope the following comments can help authors to improve the present manuscript.

COMMENTS AND SUGGESTIONS

Title

Only data pre-processing is presented, not analysis. Please correct it. In fact, authors said at the Discussion and Conclusion section that much more processing and analysis is necessary.

Abstract

Main conclusions must be include in the abstract.

1. Introduction.

- References 1 to 5 are not correctly cited at the text. Please, revise.

- Reference 2 is a web. Try to include scientific published works.

- L17. “As Elizabeth says…”. I do not understand what authors want to say.

- Figure 1. Enlarge lettering size, include a scale, and explain at the footnote what means the coordinates and how it are used. Q presume that many readers of this journal are not familiarised with this kind of geolocation.

2. Mars Rover Penetrating Radar

- Figure 2. (a) and (b) lettering must be enlarged.

- L75. First apparition of acronyms, they must be explain: HH, HV, VH and VV.

- L70. “… resolution of 1 m” and then, in L80. “… resolution of a few meters”. I feel the second is the correct one. Please, homogenise.

3. Data pre-processing

- Figure 3. As before, FFT first apparition, complete name.

- An introductory paragraph is necessary in order to explain why three levels of data products are defined.

I urge the authors not to view this solely as a critique but a chance to improve this paper to convey their ideas and data in the best possible and most accurate manner.

Warm regards

Reviewer 2 Report

The authors of the manuscript introduce the characteristics of the Mars Rover Penetrating Radar onboard Tianwen-1, including the basic information, three working modes, different levels of data products, analysis of self-check signals, calibration of Low-frequency channel data. This paper is an overview of the payload and the data. Although, more interesting results and thorough analysis are expected.  

Here are some aspects that need adjustments or clarifications:

1. Line 5, analyses should be analyze.

2. Abbreviation "RoPeR" shall not be appeared in Key words.

3. Paragraph after equation (1),  LF_Data_2C_result, LF_Data_2B_result should use italic. Same for (2)

4. The manuscript should be modified by native English speakers.

Round 2

Reviewer 1 Report

Dear authors,

All suggestions and comments haveb een taken into account. The manuscript is ready for publishing at the present form.

Regards,

Reviewer 2 Report

The authors revised the manuscript according to the suggestions. With more data obtained in the future, more interesting results and thorough analysis are expected.